# Who is meeting the strengthening physical activity guidelines by definition: A cross-sectional study of 253 423 English adults?

Gavin R. H. Sandercock[1]*, Jason Moran[1], Daniel D. Cohen[2]

1 School of Sport Rehabilitation & Exercise Science, University of Essex, Colchester, United Kingdom,
2 Masira Research Institute, Faculty of Health Sciences, Universidad de Santander (UDES), Bucaramanga, Colombia

* gavins@essex.ac.uk

**Data Availability Statement:** All data used in this study and the full Active Lives Survey Data Collection are available to users registered with the UK Data Service: https://www.ukdataservice.ac.uk/

## Abstract

The current UK physical activity guidelines recommend that adults aged 19 to 65 years perform activity to strengthen muscle and bone a minimum of twice weekly. The number of adults meeting strengthening activity guidelines is lower than for aerobic activity, but estimates vary between studies partly due to differences in how muscle-strengthening activity is defined. We aimed to provide estimates for strengthening activity prevalence in English adults based on a nationally representative sample of n = 253,423 18-65-year-olds. We attempted to quantify the variation in estimates attributable to differences in the way strengthening activity is defined. Finally, we aim to provide a brief descriptive epidemiology of the factors associated with strengthening activity. Adults met guidelines for aerobic activity if they reported the activity equivalent to >150 min/week moderate-intensity exercise. Respondents met strengthening guidelines if they reported at least two bouts per week of strengthening activity. We defined strengthening activity, first, according to criteria used in the Health Survey for England (HSE). Second, we counted bouts of strengthening activities for which we could find evidence of health-related benefits (Evidence). Third, we included bouts of strengthening activity as defined in current UK physical activity guidelines (Guideline). Two-thirds (67%) of adults met guidelines for aerobic activity (69% of men, 65% of women). Less than one-third (29% of men and 24% of women) met guidelines for the HSE definition of strengthening activity. Under the Evidence definition, 16% of men and 9% of women met strengthening guidelines. Using the most-stringent definition (Guideline) just 7.3% of men and 4.1% of women achieved the recommendations for strengthening activity. We found females and older adults (50–65 years) were less likely to meet guidelines for aerobic, strengthening, and combined aerobic plus strengthening activity. The prevalence of meeting activity guidelines was lower in adults from more deprived areas (compared with the least deprived); Adults with lower academic qualifications (Level 1) were less likely to meet activity guidelines than those educated to Level 4 (Degree Level) or higher. Having a limiting disability was associated with a lower prevalence of meeting activity guidelines. Associations between socio-demographic measures and the prevalence of adults meeting activity guidelines were stronger for strengthening activity than for aerobic 51(or combined aerobic plus strengthening) activity Compared with aerobic activity, fewer adults engage in

Commercial use of the data requires approval from the data owner or their nominee. The UK Data Service will contact you. Academic Institutional Employees can access the UKDA but are required to register, they can then login to add data to their account after which time they will be able to download these data. The data used in this study are safeguarded but available via the UK Data Service and can be obtained by request via the UK Data Archive https://www.data-archive.ac.uk/find/ Data Accession Numbers 1. Active Lives Survey, Year 1 (2016-2017) - SN-8391-1 2. Active Lives Survey, Year 2 (2017-2018) - SN-8651-1 DOIs and Persistent identifiers: 1. Sport England. Active Lives Survey, 2016-2017. [data collection]. UK Data Service, 2021 [Accessed 2 January 2022]. Available from: DOI: 10.5255/UKDA-SN-8391-1 2. Sport England. Active Lives Survey, 2017-2018. [data collection]. UK Data Service, 2021 [Accessed 2 January 2022]. Available from: DOI: 10.5255/UKDA-SN-8651-1 Further help Datasets in the main UKDA collection are usually made available in three standard formats – SPSS, Stata and TAB (tab-delimited). For information on these formats, please visit what download and data formats are available here: https://www.ukdataservice.ac.uk The authors had no special privileges in obtaining these data.

**Funding:** The author(s) received no specific funding for this work.

**Competing interests:** The authors have declared that no competing interests exist.

strengthening activity regardless of how it is defined. The range in estimates for how many adults meet strengthening activity guidelines can be explained by variations in the definition of 'strengthening' that are used and the specific sports or activities identified as strengthening exercise. When strengthening activity is included, the proportion of English adults meeting current physical activity guidelines could be as high as 1 in 3 but possibly as low as just 1 in 20. A harmonized definition of strengthening activity, that is aligned with physical activity guidelines, is required to provide realistic and comparable prevalence estimates.

## Introduction

The 2011 UK physical activity guidelines were the first to recommend at least twice-weekly bouts of strengthening activity as part of at least 150 minutes of moderate-to-vigorous physical aerobic activity per week [1]. These recommendations were based on high-quality evidence for the health benefits of muscle-strengthening activity which are independent of, and additive to, those of aerobic physical activity [2, 3].

The updated physical activity guidelines for UK adults [1] state that *"adults should undertake activities which increase or maintain muscle strength (resistance training)"*. Further description of strength activities suggests they should target upper- and lower-body muscle groups [and] comprise movements repeated to fatigue or failure'. Examples provided include *'bodyweight exercise, free weights, resistance machines or elastic (resistance) bands'*

Within the scientific literature and in public health messaging, there remains a *'preferential emphasis'* on aerobic rather than strength activity guidelines [4]. Strengthening activity is often overlooked in studies of physical activity [5] wherein adults accruing 150 weekly minutes of moderate-intensity activity are deemed to be 'meeting guidelines' or are considered 'physically active' [6–8]. This discounting of muscle-strengthening activity is acutely evident in studies reporting physical activity from accelerometers [6, 9, 10].

Excluding strengthening activities can lead to the misreporting of the population prevalence of adults who meet physical activity guidelines [11–13]. Studies including both aerobic and strengthening activities show that fewer adults meet the current physical activity guidelines [14] but estimates of how many adults meet these guidelines vary greatly. For example, Bennie et al. [15] reported that 15% of Australians met strengthening activity guidelines with just 10% meeting the recommendations for strength and aerobic activity. The CDC estimated that 20.6% of US adults met both the aerobic and strengthening guidelines in 2011 [16] while analysis of NHANES data indicates that 18–24% of US adults met strength and aerobic guidelines [17].

Using data from the Scottish Health Survey, Strain *et al*. [18] reported that 31% of men and 24% of women achieved the recommended strengthening activity guidelines. Applying the same classification criteria to data from the Health Survey for England (HSE) in 2012, Scholes [19] reported that 34% of men and 24% of women met the strengthening activity guidelines. Estimates from the 2016 HSE suggest that 31% of men and 23% of women met strength and aerobic activity guidelines.

Bennie *et al*. [14] provide what is probably the most accurate and, certainly, the most recent epidemiological description of strengthening activity in UK adults. In a Europe-wide study of strengthening activity [14], these researchers assessed responses to an item concerning weekly engagement in: "... *physical activities specifically designed to strengthen your muscles such as doing resistance training or strength exercises*". Using this definition of strengthening activity,

congruent with the description and examples provided by the UK CMO [1], less than 20% of the 20,000 UK adults surveyed met current guidelines for strengthening activity. The number of people meeting the combined aerobic and strengthening activity guidelines was not reported in this case.

Differences in reported estimates of how many adults meet the outlined guidelines likely stem from variations in how authors define strengthening activities and, therefore, the activities that 'count' toward the twice-weekly bouts recommended. For example, estimates from NHANES data are derived from an item that describes *'activities to strengthen your muscles such as lifting weights or doing calisthenics*. This item, however, prompts respondents to *'include previously mentioned aerobic activities like muscle strengthening'* thus calling into question the validity of the measure. Using a similar item but prompting respondents to *discount aerobic activities such as*: *walking*, *running*, *cycling'* the proportion meeting the recommended guidelines was just 6% [20], substantially lower than the NHANES estimate of 20% [16].

In England, engagement in muscle-strengthening activities is considered at the national level through a subjective, self-reported metric within the Health Survey for England. However, closer inspection of these data suggest that the HSE definition does not adequately differentiate muscle-strengthening activity from aerobic physical activity. The 2016 Survey showed that 43% of men and 32% of women met aerobic activity guidelines while 34% of men and 24% of women achieved the recommended level of strengthening activity. The latter figures are almost identical to the number of adults meeting both aerobic and strength guidelines (33% of men and 23% of women) suggesting that there is a substantial overlap between activities counted as aerobic and those considered as strengthening [19].

Physical activity is a minor constituent of the HSE. Indeed, the in-depth and detailed nature of the survey itself necessitates the recruitment of a relatively small (albeit nationally representative) sample of around 6000 adults. Evidence from the HSE was identified and included in a rapid review of the evidence before the 2019 update of UK Physical Activity Guidelines produced by Public Health England (PHE) [21]. The PHE review also acknowledged the much larger and more detailed assessments of sports and physical activity provided by The Active Lives Survey (ALS) but stated that it provided no assessment of muscle-strengthening activity [21]. While it is true that no summative metric of strengthening activity is routinely reported from the Active Lives Survey data, the survey assesses the frequency, duration, and intensity of participation across an exhaustive list of sports and activities in annual rolling samples of over 200,000 English adults. The Active Lives Survey also includes details of all activities listed in the HSE as well as detailed participation data based on an extensive menu of muscle-strengthening activities, classes, and sports that are not routinely included in most other health surveys.

We aimed to estimate the proportion of English adults meeting the recommended guidelines for strengthening activity and combined aerobic and strengthening activity using a nationally representative sample of adult respondents to the ALS. To assess how definitions of strengthening activity influenced estimates of prevalence, we sought to provide multiple estimates based on different definitions of strengthening activity alone and, also, in combination with a single definition of aerobic activity. A further objective was to identify factors associated with meeting guidelines for aerobic activity and strengthening activity alone and in combination.

## Methods

The Active Lives Survey was established in November 2015 and provides a world-leading approach to gathering data on how persons aged 16 and over in England engage with sport

and physical activity. The Active Lives Survey is the most comprehensive national survey of sports participation and physical activity for English adults. The overall sample size is around 175,000 people for each survey with a minimum annual sample size of 500 persons within each English local authority. The sample is randomly selected from the Royal Mail's Postal Address File and provides a sample representative of the English population across key demographic variablessuch as age, geographic spread, and levels of deprivation. The survey is performed by IPSOS-MORI which provides full details of survey development and sampling strategy [22]. Briefly, the survey is a push-to-web design whereby participants are informed of selection by mail and asked to complete the survey online. The survey is distributed in monthly waves with prior response rates allowed to influence the sampling of subsequent waves to ensure a nationally representative sample is obtained [22].

Persons aged >16 years within households in England are considered in the sampling strategy which stipulates a maximum of two respondents per household. The sampling frames and targets are intended to obtain responses from a nationally representative sample from diverse demographic and geographic areas rather than to satisfy any specific statistical query or research question. Respondents are provided with three reminders to complete the survey online and can claim a £5 shopping voucher for their participation. The third reminder also includes a hard-copy-version of the survey which can be returned by free-post to reduce any potential bias due to non-response from the 10% of UK households without internet access [22]. Respondents were informed that their replies would be used to help provide better services and consent for use of data in any secondary analysis was implied by submitting the completed questionnaire. The overall response rate for the survey across all waves reported in this study was 19% [22] which is within the normal range for large-scale surveys of this type [23].

We analysed data collected from the Active Lives Survey for the period 2015–2017 [24, 25] comprising responses from $n$ = 401 465 adults (age 16–95 years). The data were downloaded from the UK Data Archive where they are publicly available and where further technical information and detailed methodology can be obtained [24, 25]. No specific application for ethical approval was required to undertake this secondary analysis of publicly available data.

The updated methodology includes an exhaustive menu of sports, activities, exercise classes and active leisure-time pursuits, Respondents select activities in which they have participated within the past 12 months and specifically within the past four weeks (28 days). Survey routing then prompts further questions to assess the frequency intensity and duration of each activity. There are multiple items on strength- and power-based sports such as weightlifting alongside numerous items relating to resistance exercise such as kettlebell classes, circuit training, resistance machines and sessions using free weights or bodyweight resistive loads.

From the initial sample of 401,465, we removed 132,531 who were outside the age range of the current physical activity guidelines for adults (18–65 years) or who had missing values for their age or sex. Of the remaining 268,934 respondents we excluded 16,246 with MEMS> 2520 min/week; equivalent to six hours of moderate-intensity activity (MPA) or three hours of vigorous-intensity.

The Active Lives datasets include the Moderate Equivalent Minutes (MEMS) spent engaged in each activity. MEMS combines moderate physical activity (MPA) and vigorous physical activity (VPA) into a single variable. MPA and VPA are calculated by multiplying number of bouts for each activity in the past 28-days by usual bout duration with one minute VPA assumed to be the equivalent to two minutes MPA. MPA (min) and 2 x VPA (min) are combined to produce 28-day MEMS value for each activity which is then divided by four to give weekly minutes equivalent minutes of MPA (min/week). MEMS values for all activities are summed to provide an estimate of total physical activity (MEMS_ALL). Respondents are classified as physically active if MEMS_ALL is >150 min/week. IPSOS-MORI [22] and Sport

England [26] provide a complete description of the survey design, variable derivation, data cleaning and methods to minimize double-counting of activities in technical reports accompanying the data [22] available via the UK Data Archive.

From the initial sample of 401,465, we removed 132,531 who were outside the age range of the current physical activity guidelines for adults (18–65 years) or who had missing values for age or sex. Of the remaining 268,934 respondents we excluded 16,246 with MEMS> 2520 min/week equivalent to six hours of moderate-intensity activity (MPA) or three hours of vigorous-intensity physical activity (VPA) every day of the week. The initial sample of 253,423 adults was included in the primary analysis.

## Defining strengthening activity

Historically, guidelines have not included a minimum duration or required intensity for bouts of strengthening activities [1, 27–30]. Active Lives provides information on the bouts reported for every activity over the last 28 days. We summed bouts for all activities meeting each of our definitions for muscle-strengthening activity reported in the previous 28 days and divided this figure by four to provide a figure for weekly bouts. We classified respondents as meeting the strengthening guidelines if they reported ≥2 weekly bouts of muscle-strengthening. The three definitions of strengthening activity used and a list of activities included within each definition are shown in S1 Table. Herein these are referred to as follows:

The **HSE** estimate included activities described as strengthening activity in reports based on the Scottish Health Survey and Health Survey for England [18, 19].

The **Evidence** estimate included activities for which there was evidence of health benefits available in peer-reviewed studies and reviews [4, 21, 31, 32].

The **Guideline** estimate included only those activities defined or described in the 2019 UK physical activity guidelines for adults [1]:

We then calculated the percentage of adults meeting recommendations for strengthening and aerobic and strengthening activity using each definition. We examined differences in the proportion of adults meeting current physical activity recommendations.

The sociodemographic characteristics obtained were sex, age, (in 15-year bands) and quintiles of area-level deprivation (Index of Multiple Deprivation). We collapsed education to form four categories based on the highest educational qualification held. To do this we included students currently in higher education and adults already awarded degree-level (Level 4) qualifications. We combined Level 1 qualifications and no-qualification categories and included 'other' qualifications reported within the appropriate level (Level 2 and 3). The Active Lives Survey provides three-group classifications of disability status based on whether or not the individual had a limiting, non-limiting, or no disability. All were assessed using standard questionnaire items from the Active Lives Survey [33].

Of the 253,423 respondents included in the initial analysis, we excluded 3,809 due to missing sociodemographic variable values (education or disability status) thus leaving 249,614 in the secondary analysis. The raw prevalence estimates for the number of adults meeting combined aerobic and strengthening activity guidelines according to key socio-demographic variables are available in S2 Table.

To provide a descriptive epidemiology of adults meeting the current physical activity guidelines, we created binary outcome variables based on whether adults met: a) the aerobic activity guidelines, b) strengthening guidelines (based on the guideline definition) and c) the combined aerobic and strengthening activity guidelines. We calculated the likelihood of meeting (aerobic, strengthening and combined [aerobic plus strengthening]) activity guidelines and relative likelihood based on categorical sociodemographic predictors shown in Table 1. These

**Table 1. Sociodemographic characteristics of n = 275 182 English adults responding to Active Lives survey Waves 2 and 3.**

| | | Males | | Females | | All | |
|---|---|---|---|---|---|---|---|
| | | % | n = | % | n = | % | n = |
| Age | 19–34 | 32.6% | (39,640) | 32.2% | (41,183) | 32.4% | (80,823) |
| | 35–49 | 34.3% | (41,707) | 34.3% | (43,905) | 34.3% | (85,612) |
| | 50–64 | 33.2% | (40,395) | 33.5% | (42,784) | 33.3% | (83,180) |
| Deprivation (Quintile of IMD) | Least Deprived | 25.4% | (32,800) | 24.6% | (31,604) | 25.0% | (64,404) |
| | 2nd Least Deprived | 21.7% | (27,347) | 21.6% | (27,601) | 21.7% | (54,166) |
| | Median Quintile | 18.5% | (21,997) | 18.3% | (23,541) | 18.5% | (46,179) |
| | 2nd Most Deprived | 17.7% | (20,042) | 17.9% | (22,066) | 17.7% | (44,182) |
| | Most Deprived | 17.1% | (19,718) | 17.2% | (21,101) | 17.1% | (42,684) |
| Education (Highest Qualification) | Level 4 | 42.0% | (55,120) | 41.9% | (57,369) | 42.0% | (112,489) |
| | Level 3 | 21.9% | (28,659) | 23.7% | (32,438) | 22.8% | (61,097) |
| | Level 2 | 21.3% | (27,885) | 21.9% | (29,948) | 21.6% | (57,833) |
| | Level 1 | 14.8% | (19,455) | 12.5% | (17,154) | 13.7% | (36,610) |
| Occupational Status | Not Working | 4.2% | (5,615) | 5.6% | (7,863) | 4.9% | (13,478) |
| | Routine/Manual | 38.0% | (51,206) | 24.6% | (34,627) | 31.2% | (85,833) |
| | Intermediate/Study | 19.8% | (26,604) | 34.3% | (48,230) | 27.2% | (74,833) |
| | Managerial/Professional | 38.0% | (51,235) | 35.4% | (49,803) | 36.7% | (101,038) |
| Disability status | No disability | 75.2% | (96,425) | 70.2% | (93,110) | 72.7% | (189,535) |
| | Non-limiting | 13.8% | (17,677) | 14.2% | (18,782) | 14.0% | (36,460) |
| | Limiting | 11.0% | (14,100) | 15.6% | (20,744) | 13.4% | (34,844) |
| 150 min/week equivalent MPA | Meeting | 66.7% | (92,473) | 66.0% | (92,734) | 67.3% | (185,206) |
| | Not meeting | 31.3% | (42,187) | 34.0% | (47,789) | 32.7% | (89,976) |

IMD indices of multiple deprivation derived from postcode to provide area level scores at local board level or lower super output group Q1 represents the least deprived (more affluent) with five representing the most deprived areas highest Educational qualification achieved: Level 1 basic education; Level 2 Completed secondary education; Level 3- Completed further education; Level 4 attended higher education studying to Bachelors level. Disability status relates to physical disability and was self-reported and classified as no disability reported; reporting of any non-limiting disability. A physically limiting disability was defined as any condition reported to have a significant impact on tasks of daily living. 150 min/week equivalent MPA calculated as weekly minutes moderate activity (x 1) plus weekly minutes vigorous activity (x 2). A full description all measures is available in Active Lives Survey Technical Report [22]. Complete survey methodology [24] and instructions on how to access Active Lives Survey Data are available via the UK Data Archive [25]

were sex (reference: male), three age groups (reference: <35 years), deprivation (reference: least deprived quintile); socioeconomic status (referent: managerial/professional employment); highest educational qualification gained (referent: Level 4 or above) and three levels of disability status (reference category: no disability).

Due to differences in the relative frequencies of adults meeting aerobic, strength and combined guidelines we did not use binary logistic regression analyses as this method is likely to overestimate relative likelihoods for commonly occurring outcomes such as being physically active [34]. Assuming based on previous estimates that approximately 30% of adults would meet strengthening activity guidelines, binary regression would likely produce odds ratios that diverge from prevalence ratios because of the high outcome prevalence in any reference group [35]. Instead, we used Generalized Linear Models with Poisson Loglinear regression Link function and employed Robust estimates of error variance to calculate prevalence ratios (95% confidence interval). We performed three separate analyses to assess the effects of sociodemographic predictors on the prevalence ratios (PRs) for adults meeting aerobic, strengthening and combined physical activity guidelines.

## Results

After exclusions, the sample comprised 275,182 respondents aged 19–65 years (48.9% males) of whom 64.2% completed the survey electronically and 35.8% returned a hard copy version. Table 1 shows the sociodemographic characteristics of the final sample after exclusions for missing data and outlying values for MVPA. As expected in a nationally representative survey sample, the final sample retained broadly equitable proportions of participants by sex across the three age categories used and across quintiles of area-level deprivation.

Table 1 also shows the proportion of English adults aged 19 to 65 meeting the aerobic activity element of current UK guidelines. Here, meeting guidelines was defined solely as an energy expenditure equivalent to at least 150 minutes of moderate physical activity (MPA) per week but disregarded the inclusion of two bouts of strengthening activity. Using this definition, 68.7% of men and 66.0% of women met the current guidelines and, overall, 67.3% of the population were classified as physically active.

Fig 1 shows the percentage of males and females meeting current guidelines for strengthening activity only (defined as at least two sessions per week of at least 10 min duration). Using the most inclusive definition of strengthening activity, 28% of men and 25% of women met the strengthening guidelines in England. Considering only strengthening activities for which there is evidence for some health-related benefit, 16% of men and 9% of women met the strengthening guidelines. When strengthening activity was defined according to the current UK physical

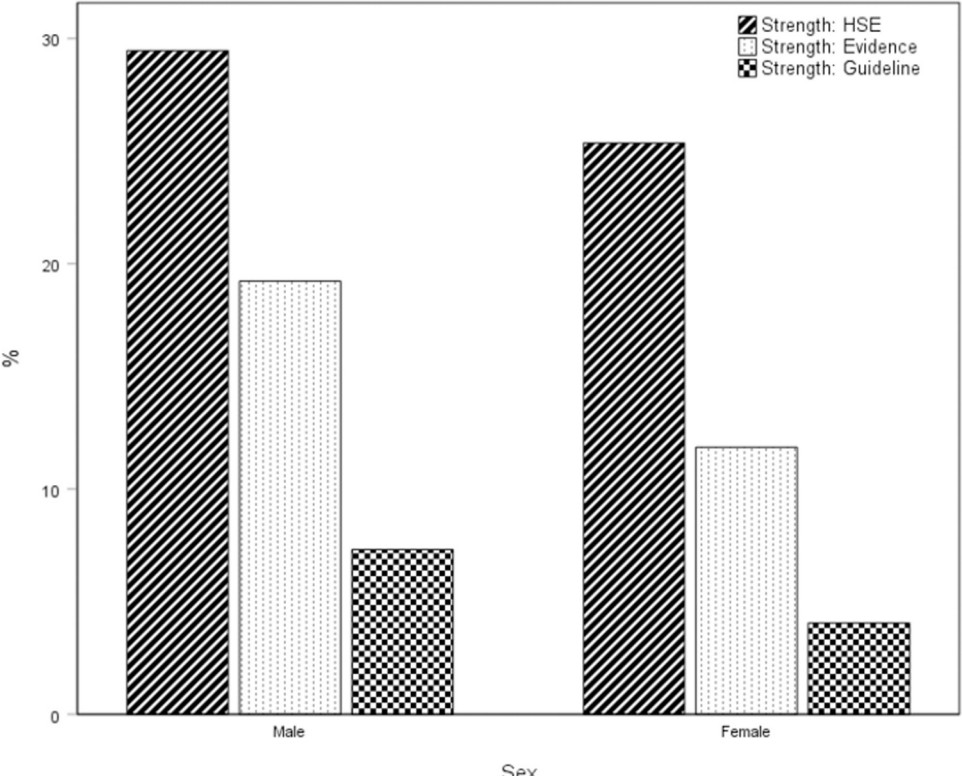

**Fig 1. The proportion of English 19-65-year-olds meeting three different interpretations of the current guidelines recommending twice-weekly strengthening activity.** *HSE* - at least two weekly sessions of strengthening activity as defined in the Health Survey for England. *(Evidence)* at least two weekly sessions of strengthening activities for which there is evidence of health benefits available within the peer-reviewed scientific literature. *Guideline*—at least two weekly sessions of strengthening activity as defined within the current UK physical activity guidelines (1). Activities included in each definition of 'Strengthening Activity' are shown in S1 Table.

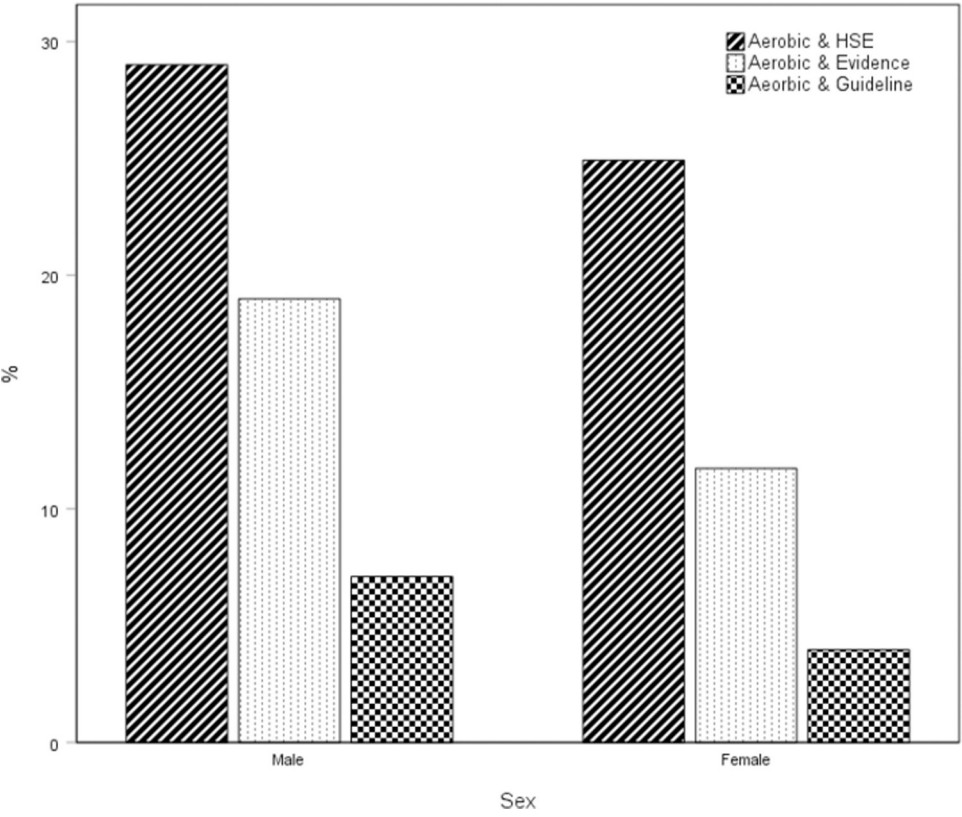

**Fig 2. The proportion of English 19-65-year-olds meeting both aerobic activity guidelines plus three different interpretations of strengthening activity.** *HSE* –aerobic activity equivalent to >150 min/week MPA including two sessions of strengthening activity as defined in the Health Survey for England. *Evidence* - 150 min/week equivalent MPA including two sessions of strengthening activities for which there is evidence of health benefits available within the peer-reviewed literature. *Guideline* - 150 min/week equivalent MPA including two sessions of strengthening activity as defined within the current UK physical activity guidelines [1]. Activities included in each definition of 'Strengthening Activity' are shown in S1 Table.

activity guidelines, overall, 5.6% of the sample met current recommendations. Again, the proportion engaging in at least two bouts of strengthening activity per week was higher in men (7.9%) than in women (4.3%)

Fig 2 shows the percentage of males and females meeting combined aerobic and strength guidelines and for aerobic-activity only 150 min/week of equivalent moderate activity. Using the most inclusive definition of strengthening activity and comparable Health Survey for England criteria 29% of men and 24% of women met the strengthening guidelines. Considering only activities for which there is evidence of strengthening-activity related, health benefits 16% of men and 9% of women were classified as physically active. When strengthening activity was defined according to the current (2019) physical activity guidelines from the UK CMO, 7.3% of men and 4.1% of women were classified as physically active.

Table 2 shows the prevalence ratios calculated from generalized linear models with Poisson regression and robust error variance to calculate prevalence ratios (95%CI). Compared with men, women were 14% less likely to meet aerobic activity guidelines (PR = 0.86 [95%CI:0.85–0.87]) and had 24% lower prevalence ratio for meeting combined aerobic plus strengthening activity. The association of sex with strengthening activity was stronger than for aerobic activity or combined activity with the prevalence ratio suggested that women were 34% less likely to meet strengthening activity guidelines than were men (PR = 0.66 [95%CI 0.65–0.68]).

**Table 2. Factors associated with meeting current UK guidelines for aerobic activity, strengthening activity and combined aerobic plus strengthening activity in a nationally representative sample of English adults aged 19–65 years.**

| | Aerobic Activity Only | | Strengthening Activity | | Aerobic Plus Strengthening Activity | |
|---|---|---|---|---|---|---|
| Sex | PR | (95%CI) | PR | (95%CI) | PR | (95%CI) |
| Male | 1.00 | Ref | 1.00 | Ref | 1.00 | Ref |
| Female | 0.86 | (0.85–0.87) | 0.66 | (0.65–0.68) | 0.76 | (0.74–0.79) |
| Age | | | | | | |
| 19–34 | 1.00 | Ref | 1.00 | Ref | 1.00 | |
| 35–49 | 1.00 | (0.99–1.01) | 0.77 | (0.74–0.80) | 0.98 | (0.97–0.99) |
| 50–64 | 0.99 | (0.98–1.01) | 0.55 | (0.52–0.58) | 0.95 | (0.94–0.97) |
| Deprivation[a] | | | | | | |
| Q1 (Least Deprived) | 1.00 | Ref | 1.00 | Ref | 1.00 | |
| Q2 | 0.98 | (0.97–1.00) | 1.02 | (0.98–1.08) | 0.99 | (0.97–1.00) |
| Q3 | 0.97 | (0.95–0.98) | 0.92 | (0.88–0.97) | 0.96 | (0.94–0.98) |
| Q4 | 0.93 | (0.92–0.94) | 0.88 | (0.83–0.93) | 0.93 | (0.91–0.96) |
| Q5 (Most Deprived) | 0.87 | (0.86–0.89) | 0.83 | (0.79–0.88) | 0.85 | (0.80–0.90) |
| Education | | | | | | |
| ≥ Level 4 | 1.00 | Ref | 1.00 | Ref | 1.00 | Ref |
| = Level 3 | 0.93 | (0.92–0.94) | 0.94 | (0.90–0.98) | 0.93 | (0.89–0.97) |
| = Level 2 | 0.86 | (0.85–0.87) | 0.79 | (0.75–0.83) | 0.78 | (0.74–0.82) |
| ≤ Level 1 | 0.75 | (0.72–0.77) | 0.62 | (0.58–0.67) | 0.73 | (0.61–0.70) |
| Disability | | | | | | |
| No Disability | 1.00 | Ref | 1.00 | Ref | 1.00 | Ref |
| Non-Limiting | 1.00 | (0.98–1.06) | 1.01 | (0.98–1.05) | 1.02 | (0.98–1.06) |
| Limiting disability | 0.82 | (0.67–0.93) | 0.66 | (0.58–0.69) | 0.80 | (0.59–0.98) |

a-Quintiles based on the Index of Multiple Deprivations (IMD) derived from postcode to provide area level scores at ward (lower super output area) level. Q1 represents the least deprived (more affluent) with five representing the most deprived areas highest Educational qualification achieved: Level 1 basic education Level 2 Completed secondary education; Level 3- Completed further education. Level 4 -College-Level educated or higher (attended higher education studying to Bachelors level or above). Disability status relates to physical disability and was self-reported and classified as noticeability or able-bodied a non-limiting disability including limiting disability if the condition was reported to have a significant impact on tasks of daily living.

Aerobic Activity Only: defined as achieving equivalent to 150 min/week moderate-intensity physical activity calculated from weekly minutes moderate-intensity activity plus 2x weekly minutes vigorous-intensity activity or any mix of the two. Aerobic Activity—based on compositive physical activity variables 'MEMS_ALL' reported in The Active Lives Survey [33], Full definitions of all measures used in this table are available in Active Lives Survey Technical Report [22]. Technical Descriptions and instructions of how to access these data are available via the UK Data Archive for Active Lives Years 1 [25] and 2 [24].

The associations between aerobic activity and aerobic and strengthening activity and age were modest. There was, however, a stronger 'dose-response relationship between meeting strengthening activity guidelines and age where the 50 to 64-year-old age group was almost half as likely to meet our definition of twice-weekly strengthening activities. Across aerobic and combined guidelines, the association with deprivation was broadly comparable. Again, associations were stronger for strengthening activity with those in the most deprived quintile being 17% less likely to meet strengthening guidelines compared with those in the least deprived quintile.

Educational status categories were based on the highesteducational qualification awarded. Respondents with ≥Level 4 qualifications (Bachelors degree or higher) were used as the reference category. Compared with Level 4-educated respondents, adults with Level 3 and Level 2 qualifications were 10%, and 15% less likely to meet aerobic activity guidelines (respectively). Respondents with educational qualifications at Level 1 or below were 25% less likely to meet aerobic activity guidelines than those with qualifications at Level 4 or above, he association of education with the prevalence of adults meeting strengthening activity guidelines was stronger than for aerobic activity. Compared with the reference group (Level 4), those educated to level 2 were 21% less likely to meet strengthening guidelines and those with qualifications equivalent to Level 1 and below 3% less likely (PR = 0.62[95%CI:0.58-.067]). to engage in twice-weekly strengthening activity.

Adults with a limiting disability had an 18% and 20% % lower prevalence ratio for meeting aerobic guidelines and combined aerobic and strengthening guidelines (respectively). The association of disability with strengthening activity was more pronounced than for aerobic activity with the prevalence ratio for meeting strengthening guidelines 34% lower compared with those reporting no disability (PR = 0.66 [95%CI:0.58–0.69]).

## Discussion

We aimed to provide estimates of the number of English adults meeting current physical activity guidelines, which comprise elements of aerobic and strengthening activity. Foster and Armstrong [4] highlighted the weaknesses in survey items previously used to assess the number of adults meeting the recommended level of strength-building activity. Also, Hillsdon [32] noted the absence of information on the frequency of participation in resistance training exercises. To address some of the methodological shortcomings of previous estimates, we used a nationally representative sample that included items assessing exercise frequency, intensity, duration and type of physical activity.

Despite the differences in design and survey items used, the present data from the Active Lives Survey agree rather well with existing estimates from The Scottish Health Survey and Health Survey for England [19], using the same definition of strengthening activity. Fig 2 shows the percentage of males and females meeting the current physical activity guidelines considering aerobic activity only and in combination with strengthening activity under each definition. In agreement with previous research [14–16, 18, 36–38], we found that, regardless of the definition used, fewer adults met the current strengthening activity recommendations compared with the number meeting aerobic activity guidelines. While more than two-thirds (67%) of adults reported the equivalent of at least 150 min/week of MPA, fewer than a quarter (23%) of the sample actually met the current UK physical activity guidelines specifying twice-weekly muscle-strengthening activity [1].

Aside from the normal requirements for intensity of physical activity, there is also the important question of what should be considered a muscle strengthening activity. By including many activities that are clearly not designed to or capable of promoting strength development, prior studies have grossly overestimated the number of adults who meet strengthening activity guidelines.

Based on the present data, the overestimations produce values approximately three times the actual number that engage in strengthening activities. The health benefits claimed for strength training are not, however, based on studies of team sports, racket sports, or the majority of activities included in the HSE definition of the activity [18, 19]. Instead, the evidence comes largely from studies using resistance training either alone or as an adjunct to other activities [4].

The HSE definition is undermined by an apparent confusion regarding the basic principles of exercise prescription: the commonly characterized FITT principles of Frequency, Intensity Time, and Type. Specifically, the definition conflates exercise intensity and exercise type (modality) by suggesting that many activities performed at a high intensity are also activities that enhance muscle strength. In this way, any activity that was considered to be *'putting muscle under tension' was included as* long as it was reported to be performed at a '*high intensity*.

The definition of 'intensity' in this context also requires clarification. During aerobic exercise, intensity refers to a constant workload directly linked to a percentage maximum–often relative to peak heart rate or $\dot{V}O_{2max}$. In relation to strength training, the concept of 'intensity' refers instead to the magnitude of the resistive load (weight lifted) expressed relative to the maximal load that could be lifted in a single effort (most commonly a percentage of a 'one repetition maximum'). The intensity of exercise (or load lifted) determines the number of repetitions that are possible for any given movement with an inverse relationship between the magnitude of the load and the number of repetitions a trainee can execute within a given set. Strength training is, therefore, commonly a high-intensity activity but high-intensity aerobic exercise is not, by default, strengthening activity [39].

Despite this observation, the HSE definition seemingly includes team (ball) sports and racket sports regardless of the limited evidence that these could be classified as strengthening activities [31]. Of 26 sports reviewed, Oja *et al.* [31] found strengthening benefits only for running, tennis and football., This is an unsurprising outcome given that to produce high forces, and to generate the necessary amount of mechanical tension for adaptation in most commonly used resistance exercises, muscle fibers must shorten slowly against a relatively heavy resistive load [40, 41].

In addition to football, aerobics, [18] and cycling [42] are two of the most prevalent forms of activity reported within the UK survey data. Despite being relatively well-investigated [31], evidence for muscle strengthening benefits in healthy adults remains inconclusive. Swimming, walking and cycling were not included in the evidence definition that suggested 15% of adults met recommendations for strengthening activity. While lower than previous UK estimates, it is noteworthy that this definition included a composite measure of 'running'. Running is the most commonly reported leisure-time activity in Active Lives Survey respondents aged 19 to 65 years but was classified as a strengthening activity based on evidence from a rapid review of literature produced by Public Health England [21]. This review, and others [4], suggested that running could exert only a small effect on muscle function. In contrast, Oja et al. [31] concluded that the evidence for benefits to muscle strengthening was inconclusive.

Regardless of the quality of evidence, running does not meet the definition of muscle strengthening activity recommended for adults within the current guidelines [1]. This activity was, therefore, omitted from the final guideline definition of strengthening activity.

When including only activities that met the description provided in current guidelines, just 5% of adults met the recommendations. This proportion is comparable to that reported for US adults using a similarly stringent definition of what constitutes strengthening activity [20] but is considerably higher than the estimate recently reported in UK adults [14]. We included strengthening activities only if performed in bouts of ten minutes or more. Stipulating a minimum bout duration reduces estimates of how many adults meet aerobic activity guidelines [43].

Alternatively, disparities may be due to methodological differences in our approach to assessing strengthening activity. Bennie *et al.* [14] assessed responses to a single item to capture all activities perceived to '*strengthen your muscles such as doing resistance training or strength exercises'*. The guideline definition was designed to capture a comparable range of

strengthening activity using a different approach; compiling all bouts reported for numerous activities selected from an exhaustive list. The similarity in prevalence rates in our study using the HSE definition suggest the Active Lives Survey can provide comparable estimates to smaller UK surveys. Given the definitive menu of activities captured, and the large representative sample provided by the Active Lives Survey, we are confident that our estimates represent the prevalence rate of strengthening activities in English adults.

To produce valid and realistic prevalence estimates of any health behaviour, the chosen outcome measure must accurately reflect an agreed definition of the behaviour. In terms of physical activity, outcome measures should reflect the behaviours described within relevant guidelines (CMO 2019). The recommendation that adults perform twice-weekly strengthening activities are largely based on evidence for the health benefits of undertaking deliberate, purposeful muscle-strengthening activity [3, 17]. Only the 'guideline' definition used here reflects the description and examples for strengthening activity provided in the 2019 UK Physical Activity Guidelines

The importance of how strengthening activity should be defined has been highlighted previously [14, 44]. Discussion of which activities are incorporated within any unified definition of 'strengthening activity' transcends mere semantics and should not be taken lightly. This is because the behaviour of interest (strengthening activity) elicits specific physiological responses such as muscular hypertrophy, increases in bone density and enhanced force producing capabilities which confer health benefits and are different to those derived from other forms of exercise [45]. Dankel et al. [2] provided an elegant illustration by comparing the prognostic power of meeting strength guidelines (behaviour) and objectively measured muscle strength (outcome). The 10-year risk of all-cause mortality in adults who met the strengthening guidelines was lower only in those with good muscle strength. In adults meeting strengthening activity guidelines but lacking (paradoxically) good muscle strength, no such benefits were observed. Dankel et al. [2] concluded that the outcome of strengthening activity rather than the behaviour is responsible for the health benefits observed.

In short, to benefit health, strengthening activities must improve strength; the 2019 update to the UK physical activity guidelines clearly describes and provides examples of just such activities. Bennie et al's [14] recent epidemiology of European adults defined strengthening activity as: '*physical activities specifically designed to strengthen your muscles*' a definition reflected in the guideline estimate used in the present study. The proportion of adults meeting physical activity guidelines that include aerobic and strengthening activity defined in this way is startlingly small at ~5% compared with less stringent definitions of strengthening (~30%) or when considering only aerobic activity. (~67%).

The use of self-report tends to overestimate individual levels of physical activity and therefore, to inflate population estimates of how many adults meet recommendations [46]. The stark contrasts between estimates produced by HSE and Guideline may indicate the latter is an overly stringent definition of strengthening activity. We acknowledge the possibility that the guideline definition estimate is conservative with just 6% of adults being classified as physically active. Simultaneously, it is reasonable to suggest the aerobic activity estimate of 67% is somewhat inflated. The derivation of hugely contrasting estimates for physical activity are not, however, without precedent [47, 48].

According to self-reported data, 54.1% of women and 59.8% of men met current physical activity recommendations. The equivalent figures for women and men using accelerometer-based MVPA, measured in 10 min bouts were just 11.7% and 16.6% respectively.

According to self-reported physical activity data from the 2008 HSE; 39% of men and 29% of women met recommendations for physical activity [49]. Analysis of objective physical activity data from accelerometers worn by a subsample of participants revealed that only 6% of

men and 4% of women met recommendations. Furthermore, only 8% of men and 10% of women who reported meeting recommendations did so when activity was measured objectively. Methodological variations prohibit direct comparison, but the latter estimates are near-identical to those reported presently. The agreement between these two very different methods could be interpreted as coincidental. Alternatively, the guideline definition may provide a more realistic estimate of the proportion of adults meeting current physical activity guidelines.

## Sociodemographic correlates

In agreement with previous studies [18, 37, 38] we found all sociodemographic measures included in this study showed more pronounced associations with strengthening activity compared with aerobic activity (or combined aerobic and strengthening activity). Compared with aerobic activity guidelines, differences in the likelihood of meeting strengthening activity guidelines were much more pronounced by sex and age [15]. Table 2 shows that women were 14% less likely than men (referent group) to meet aerobic activity guidelines (PR = 0.86 [95% CI: 0.85–0.87]). Women were, however, 34% less likely to meet strengthening activity guidelines (PR = 0.66 [95%CI: 0.65–0.68]).

Differences in how age predicted the likelihood of meeting aerobic or strengthening activity guidelines were even starker. Using 19-34-year-olds as the referent group, Table 2 shows 35-49-year-olds and 50-64-year-olds were just as likely to meet aerobic activity guidelines. When considering strengthening activity 5-49-year-olds were 23% less likely (PR = 0.77 [95% CI:0.74–0.80]) and 50-64-year-olds were 45% less likely (PR = 0.55 [95%CI:0.52–0.58]) to meet the guidelines. The association between deprivation and strengthening activity was also stronger than the influence on aerobic activity. Compared with aerobic activity, strengthening was more greatly influenced by education with stronger evidence of a negative 'dose-response' relationship between respondents' highest educational qualification (level of education) and the likelihood of them meeting the strengthening activity guidelines [14]. Self-reported health is a known correlate of strengthening activity. The differences in prevalence ratio values shown in Table 2 suggest that adults with a physically limiting disability are much less likely to meet strengthening guidelines and that the influence of disability on this likelihood is more pronounced than it is for aerobic activity.

One explanation as to why socio-demographic factors influence strengthening more than aerobic activities is accessibility. Gyms and resistance training facilities may be less accessible to adults with limiting physical disabilities [50], they may be less welcoming to women and older adults [51] or their cost may be prohibitive for those from more deprived areas or those on low incomes [52]. Level of education may also act as a proxy for economic status (and level of education is negatively associated with deprivation). In agreement with others [38], and independently from deprivation [52] we found adults with qualifications indicative of fewer years of education were less likely to meet strengthening activity guidelines. This association was more pronounced for strengthening than for aerobic activity. This could reflect better awareness of the health-related benefits and of the guidelines themselves in adults with higher academic qualifications [15, 53].

## Limitations

A number of authors have recommended identifying strengthening activities from surveys as an alternative to analysing responses to items assessing 'resistance training' as a whole [32, 36]. We did not include older adults (>65 years) who make up a large proportion of the UK population. Older adults have different physical activity habits to those aged 19 to 65 years with a much greater proportion of overall activity coming from pursuits such as walking, cycling, and

gardening. Guidelines for older adults (>65 years) include muscle-strengthening activities to promote balance and prevent falls. This encompasses a different range of activities to those recommended for adults (19 to 64 years) in whom the focus is on strengthening muscle and bone. Reviews of the evidence produced in the lead-up to the 2019 UK Guidelines often failed to discriminate between muscle strengthening exercise and activities that contribute to balance. The same issue is evident in expert panel meetings that fed into the classification of strengthening activity in UK health surveys that may have led to the inclusion of several activities that are not recognised as muscle strengthening in previous estimates. These differences in habitual activities suggest that the impact of including strengthening activity in any definition of 'meeting recommendation' would be starker in this group than in the adult data reported here. Our reasoning for not including older adults was because of differences in the definition of strengthening activity and the benefits evidenced in current physical activity guidelines.

We excluded adults reporting the equivalent of; 2520 min/week of moderate-intensity activity. This figure is the equivalent of >6 hours of moderate-intensity activity each of the week is lower than the 8-h/day cut-off used in older adults in a recent analysis of data from the Active Lives Survey [54] and maybe somewhat conservative as it is achievable if respondents partake in 3-h vigorous activity day Excluding this 6% of the sample inevitably reduce r estimate of how many adults met current physical activity recommendations. When these cases were considered in our sensitivity analysis (S3 Table) dg aerobic activity guidelines increased to 70.7%. The proportion of respondents with very high overall activity who met strengthening guidelines was 16.5%; four times higher than in less-active respondents. Including these cases increased the proportion of adults estimated to be meeting strengthening guidelines to 5.4%. The percentage of adults who met combined aerobic and strengthening guidelines also increased but remained relatively low at5.3%. To identify whether the exclusion of potential over-reporters impacted our estimates of the association between sociodemographic characteristics and the likelihood of meeting aerobic or strengthening activity guidelines we performed a sensitivity analysis; reproducing the generalized linear model shown in Table 2 when including these cases. The equivalent exponential estimates of the association are shown in S3 Table. The most obvious observation from this analysis is that the inclusion of this small minority of over-reporters makes little difference to the overall conclusions of this study.

Relatively little is known of the inherent biases in push-to-web surveys compared with online only or hard-copy only methods but Ipsos-Mori Provides a detailed account of all countermeasures employed to ensure that the Active Lives Survey provides a representative sample of the English population and an accurate representation of English adults' physical activity [22]. At 67–70% however the proportion of respondents who meet guidelines for aerobic physical activity is much higher than in other [19] or other parts of the UK [18]. The modest incentives to complete the Active Lives Survey provided by Ipos-Mori and Sport England seem unlikely to be a source of bias but the branding and source of the survey itself may well be. There is evidence that the source of a survey may bias response rates. In the Case of Active Lives, the branded source of the survey is Sport England the activity levels of respondents to a survey about 'sport' are more active than the population as a whole [23]. This again suggests the numbers reported in this study may still be overestimations of the number of English adults meeting aerobic, strengthening, and combined physical activity guidelines.

## Conclusions

Despite national and international recommendations including specific statements on the importance of physical activity to strengthen muscle and bone, these recommendations are rarely measured in national surveillance systems [44]. Including muscle-strengthening activity

by a more accurate definition in physical activity surveys greatly reduces the population prevalence of adults meeting UK guidelines. Applying the least-stringent definition reduces the estimate of how many adults are physically active from >66% to <25%. Under the most stringent definition of strengthening activity, we found around 1 in 20 adults met current guidelines, a stark contrast to the estimate that 2 in 3 adults are physically active based solely on aerobic activity.

Of fifty-eight different survey systems across seventeen countries, 85.3% provided information on frequency of muscle-strengthening activity. Assessment of muscle-strengthening varied greatly however with <1% of sources reporting the validity and reliability of assessment items used [55].

The UK boasts five surveys assessing the moderate-to-vigorous (MVPA) physical activity recommendation, with three of these assessing muscle-strengthening activity. However, methodological variations between the utilised assessment tools make comparable prevalence estimates difficult [5].

In a review published just prior to the update of UK physical activity guidelines Milton et al. [44] highlighted that muscle strengthening activity was not routinely considered when calculating national estimates of how many adults are physically active. At the time of writing (January 2022) we accessed the UK Government's 'Fingertips' portal which provides area-level information on key public health indicators including physical activity [56]. While this innovation provides users with an estimate of 'percentage of physically active adults', the estimate is based solely on weekly minutes of moderate activity. The present findings support further recommendations made by Milton et al. [44] and others [57] highlighting the need to establish which sports and activities count towards different aspects of the current physical activity guidelines [44].

## Supporting information

**S1 Table. Activities included within each definition of strengthening activity.**
(DOCX)

**S2 Table. Percentage of males and females (19–65 years) meeting three definitions of the aerobic and strengthening activity guidelines according to age, deprivation, level of education, and disability status.**
(DOCX)

**S3 Table. Comparative sensitivity analysis for factors associated with meeting current UK guidelines for aerobic activity, strengthening activity and combined aerobic plus strengthening activity in a nationally representative sample of English adults (19–65 years) including those with MVPA values >2280 min week.**
(DOCX)

**S4 Table. Comparative sensitivity analysis for predictors of aerobic and strengthening activity including aerobic activity values greater than 2280 minutes per week.**
(DOCX)

## Author Contributions

**Conceptualization:** Gavin R. H. Sandercock, Daniel D. Cohen.

**Data curation:** Gavin R. H. Sandercock.

**Methodology:** Jason Moran.

**Writing – original draft:** Gavin R. H. Sandercock, Jason Moran, Daniel D. Cohen.

**Writing – review & editing:** Jason Moran, Daniel D. Cohen.

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
