## [Decision Letter · Decision Letter 0]

21 Dec 2021

PONE-D-21-24467Who is meeting the strengthening physical activity guidelines by definition: a cross sectional study of 253423 English adultsPLOS ONE

Dear Dr. Sandercock,

Thank you for submitting your manuscript to PLOS ONE. After careful consideration, we feel that it has merit but does not fully meet PLOS ONE’s publication criteria as it currently stands. Therefore, we invite you to submit a revised version of the manuscript that addresses the points raised during the review process.

We look forward to receiving your revised manuscript.

Kind regards,

Martin Senechal, PhD

Academic Editor

PLOS ONE

2. Please note that in order to use the direct billing option the corresponding author must be affiliated with the chosen institute. Please either amend your manuscript to change the affiliation or corresponding author, or email us at plosone@plos.org with a request to remove this option.

Reviewers' comments:

Reviewer's Responses to Questions

**Comments to the Author**

1. Is the manuscript technically sound, and do the data support the conclusions?

Reviewer #1: Yes

2. Has the statistical analysis been performed appropriately and rigorously? 

Reviewer #1: Yes

3. Have the authors made all data underlying the findings in their manuscript fully available?

Reviewer #1: Yes

4. Is the manuscript presented in an intelligible fashion and written in standard English?

Reviewer #1: Yes

5. Review Comments to the Author

Reviewer #1: Reviewer comments

This study aimed to provide estimates for strengthening activity prevalence in English adults and to quantify the variation in estimates attributable to differences in the way strengthening activity is defined. The study also aimed to study the association between sociodemographic factors and the performance of strengthening activities. The study is well written and addresses an important topic, i.e. how differences in prevalence estimates can be due to differences in the definitions of a behavior or condition. The statistical methods used are appropriate for the research question.

General comments:

1. In what way are the participants contacted about participation in the Active Lives Survey (ALS)?

2. What proportion of persons who are asked to participate in the ALS decline participation? Is there any data on the non-participants of the ALS, for example regarding age and sex distribution? In what way could the reward of a £5 shopping voucher affect which groups of persons choose to participate in the ALS?

3. There is no reference to the supporting information in the text. Please add a sentence in the methods section indicating that additional information on the activities included in each definition of strengthening activities can be found in the supporting information. Instead, on line 211, there is a reference to Table 1 which does not include the definitions of strengthening activities.

4. I suggest that you consider performing a sensitivity analysis of the association between sociodemographic factors and meeting strengthening guidelines where participants reporting very high levels of aerobic physical activity are included. Would the inclusion of these individuals, who probably have a high level of physical activity, although most likely not as high as reported, change your results? Are the excluded individuals similar to the included individuals in characteristics such as age and sex?

5. Please check that all abbreviations are spelled out at first use.

Comments on specific sections of the manuscript:

Line 28: Please change ”quantifying” to ”quantify”

Line 29: The word “in” is missing

Line 42: The word ”women” is missing

Lines 277-278: Please replace “open brackets”, “close brackets” with actual brackets.

Line 353: Please replace “open brackets”, “close brackets” with actual brackets.

Lines 365-368 and 373: The meaning of ”Strengthening guidelines” and “Aerobic and combined guidelines” is not immediately apparent. Please consider explaining this more clearly.

Line 370: Please change “slightly” to “likely”.

Line 457: Please change “thae” to “that”

Lines 538-539: “more pronounced associations between strengthening activity than for either aerobic activity (or combined aerobic and strengthening activity).” The meaning of this statement is unclear. Do you mean the association between the sociodemographic measures and level of strengthening activity? Please consider rephrasing this statement.

Lines 539-540: “Differences in strengthening activity were much more pronounced in middle-aged adults and women”. Please elaborate. What was the direction of the differences mentioned and what was the comparison group?

Line 550: Please change “at” to “as”

Lines 550-553: If possible, please provide references supporting these theories. If there are no previous studies of the accessibility/perceived accessibility of gyms and resistance training facilities for women, older persons, and adults with disabilities, please state this.

Line 615: “national prevalence estimates”. Please state what prevalence was estimated.

6. PLOS authors have the option to publish the peer review history of their article (what does this mean?). If published, this will include your full peer review and any attached files.

Reviewer #1: No

---

## [Author Response · Author response to Decision Letter 0]

2 Mar 2022

2. Please note that to use the direct billing option the corresponding author must be affiliated with the chosen institute. Please either amend your manuscript to change the affiliation or corresponding author or email us at plosone@plos.org with a request to remove this option.

This is now the case

These details are now included in our cover letter

We have taken the second of these options – we do not have a license to share Sport England’s data per se but we have provided stable URLs and DOIs to enable researchers to access the full raw data via the UK data repository. 

Important: If there are ethical or legal restrictions to sharing your data publicly, please explain these restrictions in detail. Please see our guidelines for more information on what we consider unacceptable restrictions to publicly sharing data: http://journals.plos.org/plosone/s/data-availability#loc-unacceptable-data-access-restrictions. Note that it is not acceptable for the authors to be the sole named individuals responsible for ensuring data access. We will update your Data Availability statement to reflect the information you provide in your cover letter.

Reviewers' comments:

Reviewer's Responses to Questions

Comments to the Author

1. Is the manuscript technically sound, and do the data support the conclusions?

Reviewer #1: Yes

2. Has the statistical analysis been performed appropriately and rigorously?

Reviewer #1: Yes

3. Have the authors made all data underlying the findings in their manuscript fully available?

Reviewer #1: Yes

4. Is the manuscript presented in an intelligible fashion and written in standard English?

Reviewer #1: Yes

5. Review Comments to the Author

In the markup version of our revised manuscript, you will find responses to each of the comments below. wear these comments relate to minor changes to the wording they are shown in the document as track changes. in the case of additions or where it was necessary to undertake wholesale rewording of sections or paragraphs changes are shown in red rather than track changes to ensure readability.

The same system in our responses hello where responses to comments, where explanation is required, are shown in red and minor modifications based on your helpful comments as track changes. 

Reviewer #1: Reviewer comments

This study aimed to provide estimates for strengthening activity prevalence in English adults and to quantify the variation in estimates attributable to differences in the way strengthening activity is defined. The study also aimed to study the association between sociodemographic factors and the performance of strengthening activities. The study is well written and addresses an important topic, i.e. how differences in prevalence estimates can be due to differences in the definitions of a behavior or condition. The statistical methods used are appropriate for the research question.

General comments:

1. In what way are the participants contacted about participation in the Active Lives Survey (ALS)?

We have included a section in the methods explaining that this is a push-to-web survey whereby respondents Receive three invitations to take part in the study two of which asked him to fill it in online the third of which asked him to fill it in online again and then provides them with a hard copy of the study which they can return by free post.

2. What proportion of persons who are asked to participate in the ALS decline participation? Is there any data on the non-participants of the ALS, for example regarding age and sex distribution? In what way could the reward of a £5 shopping voucher affect which groups of persons choose to participate in the ALS?

We have included the response rate (19%) 

As per the effect of the £5 voucher, this answer we have unavoidably avoided as we do not know the answer. we have however included information about how the survey is sent out in waves to ensure that it is representative of the English population in the limitations section. we hope that this together with the added detail that we have provided in the methods about how the survey is administered is satisfactory. 

3. There is no reference to the supporting information in the text. Please add a sentence in the methods section indicating that additional information on the activities included in each definition of strengthening activities can be found in the supporting information. Instead, on line 211, there is a reference to Table 1 which does not include the definitions of strengthening activities.

I think apologies are in order here this should have been table S1 and we have now made reference to all supporting material including this table this one table S2 and the new table S3 which we have included, based on your suggestion to perform a sensitivity analysis. For simplicity and brevity, this takes the form of a simple repeating of the original analysis with the same reporting standards. Table S3, therefore, contains the same exponential parameters estimates as table 2 with the previously excluded cases included in the analysis.

REFERENCE to Table S1 IS NOW MADE IN LINE 222

REFERENCE to Table S2 IS Now made in LINE 250

We have now included Table S3-as per your suggestions of a sensitivity analysis (see responses below).

REFERENCE to Table S3 is now made in lin 618

4. I suggest that you consider performing a sensitivity analysis of the association between sociodemographic factors and meeting strengthening guidelines where participants reporting very high levels of aerobic physical activity are included. Would the inclusion of these individuals, who probably have a high level of physical activity, although most likely not as high as reported, change your results? Are the excluded individuals similar to the included individuals in characteristics such as age and sex? 

We have taken up your suggestion to perform a sensitivity analysis. For simplicity and brevity this takes the form of repeating the original analysis, we have used the same reporting standards and included the results in the Supporting Information (Table S3) which contains the same exponential parameter estimates as table 2 with the previously excluded cases included in the analysis.

5. Please check that all abbreviations are spelled out at first use. Now Checked and shown as track changes in mark up.

Comments on specific sections of the manuscript:

Line 28: Please change ”quantifying” to ”quantify” changed to ‘quantify’

Line 29: The word “in” is missing changed to: ‘differences ‘in’ the way’ 

Line 42: The word ”women” is missing ‘of women’ now added

Lines 277-278: Please replace “open brackets”, “close brackets” with actual brackets. Apologies for this – it seems some coding from MacBook to Windows has taken ‘brackets’ as a word? Now changed.

Line 353: Please replace “open brackets”, “close brackets” with actual brackets. Again Apologies for this – it seems some coding from MacBook to Windows has taken ‘brackets’ as a word? Now changed.

Lines 365-368 and 373: The meaning of ”Strengthening guidelines” changed to ‘meeting strengthening acvtivity guidelines’ to fit with the rest of the text and “Aerobic and combined guidelines” is not immediately apparent. Please consider explaining this more clearly.

Line 370: Please change “slightly” to “likely”. Changed slightly to ‘likely’

Line 457: Please change “thae” to “that” Changed from ‘that thae’ to ‘that the’

Lines 538-539: “more pronounced associations between strengthening activity than for either aerobic activity (or combined aerobic and strengthening activity).” The meaning of this statement is unclear. Do you mean the association between the sociodemographic measures and level of strengthening activity? Yes, that’s what we meant – now changed Please consider rephrasing this statement.

Lines 539-540: “Differences in strengthening activity were much more pronounced in middle-aged adults and women”. Please elaborate. What was the direction of the differences mentioned and what was the comparison group? As above – this section has now been changed to provide a more expansive explanation of the direction of association observed including details of the referent groups in each category. 

Line 550: Please change “at” to “as” now changed 

Lines 550-553: If possible, please provide references supporting these theories. If there are no previous studies of the accessibility/perceived accessibility of gyms and resistance training facilities for women, older persons, and adults with disabilities, please state this. There are a number of studies in this area investigating barriers to participation in different modes of sports and exercise – we have now included three references in relation to disability, gender/age and socio-ecomonimic status. 

Line 615: “national prevalence estimates”. Please state what prevalence was estimated.Wording changed

---

## [Editor Report · Decision Letter 1]

6 Apr 2022

Who is meeting the strengthening physical activity guidelines by definition: a cross sectional study of 253423 English adults

PONE-D-21-24467R1

Dear Dr. Sandercock,

We’re pleased to inform you that your manuscript has been judged scientifically suitable for publication and will be formally accepted for publication once it meets all outstanding technical requirements.

Kind regards,

Martin Senechal, PhD

Academic Editor

PLOS ONE
---

## [Editor Report · Acceptance letter]

11 Apr 2022

PONE-D-21-24467R1 

Who is meeting the strengthening physical activity guidelines by definition: a cross-sectional study of 253 423 English adults? 

Dear Dr. Sandercock:

I'm pleased to inform you that your manuscript has been deemed suitable for publication in PLOS ONE. Congratulations! Your manuscript is now with our production department. 

Kind regards, 

on behalf of

Dr. Martin Senechal 

Academic Editor

PLOS ONE